# Biocatalytic Preparation of Chloroindanol Derivatives. Antifungal Activity and Detoxification by the Phytopathogenic Fungus *Botrytis cinerea*

**DOI:** 10.3390/plants9121648

**Published:** 2020-11-25

**Authors:** Cristina Pinedo-Rivilla, Javier Moraga, Guillermo Pérez-Sasián, Alba Peña-Hernández, Isidro G. Collado, Josefina Aleu

**Affiliations:** 1Departamento de Química Orgánica, Facultad de Ciencias, Universidad de Cádiz, Puerto Real, 11510 Cádiz, Spain; cristina.pinedo@uca.es (C.P.-R.); javier.moraga@uca.es (J.M.); guille.perezsasian@gmail.com (G.P.-S.); albaph93@gmail.com (A.P.-H.); isidro.gonzalez@uca.es (I.G.C.); 2Departamento de Biomedicina, Biotecnología y Salud Pública, Área de Microbiología, Facultad de Ciencias, Universidad de Cádiz, Puerto Real, 11510 Cádiz, Spain

**Keywords:** baker’s yeast, lipase, biotransformation, *Botrytis cinerea*, chloroindanol derivatives

## Abstract

Indanols are a family of chemical compounds that have been widely studied due to their broad range of biological activity. They are also important intermediates used as synthetic precursors to other products with important applications in pharmacology. Enantiomerically pure chloroindanol derivatives exhibiting antifungal activity against the phytopathogenic fungus *Botrytis cinerea* were prepared using biocatalytic methods. As a result of the biotransformation of racemic 6-chloroindanol (**1**) and 5-chloroindanol (**2**) by the fungus *B. cinerea*, the compounds *anti*-(+)-6-chloroindan-1,2-diol (*anti*-(+)-**7**), *anti*-(+)-5-chloroindan-1,3-diol (*anti*-(+)-**8**), *syn*-(+)-5-chloroindan-1,3-diol (*syn*-(+)-**8**), *syn*-(-)-5-chloroindan-1,3-diol (*syn*-(-)-**8**), and *anti*-(+)-5-chloroindan-1,2-diol (*anti*-(+)-**9**) were isolated for the first time. These products were characterized by spectroscopic techniques and their enantiomeric excesses studied by chromatographic techniques. The results obtained in the biotransformation seem to suggest that the fungus *B. cinerea* uses oxidation reactions as a detoxification mechanism.

## 1. Introduction

Optically active indanol derivatives are important precursors to other biologically active compounds which, in turn, play a key role in the development of pharmaceuticals and medicinal chemistry [1]. For instance, (1*S*, 2*R*)-1-aminoindan-2-ol has attracted interest due to its usefulness in the synthesis of indinavir^®^, a potent protease inhibitor of the human immunodeficiency virus (HIV) [2]. Indanols have also been reported to have anti-inflammatory [3], antifungal [4], and other interesting biological properties [5].

The asymmetric synthesis of these molecules has been widely studied using chiral chemical reagents, especially organocatalysts [6,7,8,9]. However, in recent years there has been a pressing need for alternative, cleaner chemistry that is more resource-efficient and produces less waste. Thus, biocatalytic methods form part of sustainable green technology [10].

Biological methods, based on enzymes and microorganisms, are capable of catalyzing reactions and have resulted in good yields and enantiomeric excesses [11]. These catalysts have advantages over chemical methods such as the low cost of materials, little environmental pollution, and recent research which has been steadily increasing the number of available enzymes and microorganisms.

*Botrytis* species are widely distributed worldwide causing serious damage to a wide variety of ornamental plants and crops. One of the most known species is *B. cinerea* which causes the disease known as grey mold affecting approximately 596 genera of vascular plants accounting for over 1400 ornamental and agriculturally important plant species [12,13]. This disease has serious economic consequences for agriculture damaging important crops around the world such as grapes, apricots, olives, and strawberries. Our research group has studied the metabolism and genetics of this fungus [14,15,16,17] and, in recent years, has focused on designing new more selective fungicides with fewer negative environmental consequences and which prevent the emergence of resistance by microorganisms [18].

Thus, we identify several chloroindane derivatives which exhibited antifungal properties against *B. cinerea* after screening compounds analogous to various phytoalexins.

In this paper, we report on the biocatalytic preparation of the enantiomers of 6-chloroindanol (**1**) and 5-chloroindanol (**2**) and their antifungal properties against *B. cinerea* UCA992.

Moreover, the chloroindanols obtained were biotransformed by *B. cinerea* to study a possible detoxification mechanism of the fungus, since it is known that several phytopathogenic fungi detoxify phytoalexins producing less toxic compounds than the substrates.

## 2. Results and Discussion

Racemic alcohols **1** and **2** were obtained from the commercially available ketones 6-chloroindanone (**3**) and 5-chloroindanone (**4**) by means of a chemical reduction as described in the Experimental Section. The racemic alcohols were then acetylated to obtain racemic acetates 6-chloroindanyl acetate (**5**) and 5-chloroindanyl acetate (**6**). Alcohols were identified by comparing NMR data from the literature for both compounds with those obtained in our reduction reactions [6,9,19]. Acetate derivatives are described here for the first time.

Enantiomerically pure alcohols derived from **1** and **2** were prepared using baker’s yeast and lipases as biocatalysts. These two methods were chosen based on abundant availability, ease of use, and the environmentally friendly conditions employed in the reactions.

### 2.1. Baker’s Yeast Reduction

Baker’s yeast is commonly used in the successful reduction of a variety of carbonyl compounds into optically active alcohols with *S*-configurations [4,20,21,22,23].

A transformation product was isolated from each ketone following incubation of **3** and **4** in fermenting baker’s yeast according to the conventional procedure (see Experimental Section): (*S*)-(+)-6-chloroindanol ((*S*)-(**1**)) (66% *ee*, 3%) and (*S*)-(+)-5-chloroindanol ((*S*)-**2**) (>99% *ee,* 40%) at 168 and 148 h of reaction, respectively (Scheme 1). The best results are obtained with 5-chloroindanone (**4**), suggesting that the position of the heteroatom could affect the interaction of the compound with the active site of the enzyme responsible for the reduction reaction. The *S*-configuration was determined by comparing the specific rotation value with data from the literature [6,9,19].

### 2.2. Lipase-Mediated Transformations

Lipases are a group of enzymes well known for their activity as acetylation agents leading to *R*-enantiomers with good yields and enantiomeric excesses, in accordance with Kazlauskas’ empirical rule for secondary alcohols [4,24]. We, therefore, focused on the kinetic resolution of racemic alcohols **1** and **2** obtained as described in the Experimental Section (Scheme 2).

Starting material **1** or **2** were added to different media with vinyl acetate as an acyl donor and *tert*-butyl methyl ether as the organic solvent and two different lipases: PPL (porcine pancreas lipase) and CRL (*Candida rugosa* lipase) (Scheme 2). The lipase CRL seemed to be more efficient for the enzymatic acetylation of these substrates, simultaneously affording the highest enantioselectivities and yields. PPL was rejected as a possible mediator for substrate **2** due to the low selectivity and slow conversions observed. The results are summarized in Table 1.

After hydrolysis of acetate derivatives **5** and **6** with potassium hydroxide in methanol, the enantiopure alcohols were identified as (*R*)-(-)-6-chloroindanol ((*R*)-**1**), with >99% *ee* and a conversion of 17% using PPL, and 95% *ee* and a conversion of 50% using CRL. (*R*)-(-)-5-chloroindanol ((*R*)-**2**) was purified with >99% *ee* and a conversion of 50% with CRL as lipase. The best results were obtained with alcohol **2** and the lipase CRL.

### 2.3. Antifungal Assays

Once we obtained the enantiopure compounds using biological methods, we planned the study of their antifungal properties against the phytopathogenic fungus *B. cinerea* UCA992 using the “poisoned food” technique [25]. Since all compounds tested exhibited a total inhibition of the fungus growth at 200 µg/mL, results of the bioassays are shown at 100 µg/mL to compare the antifungal properties of the compounds (Appendix A).

In percentage terms, the alcohol (*S*)-(+)-5-chloroindanol ((*S*)-**2**) exhibited the maximum growth inhibition against *B. cinerea* UCA992 followed by ketone **3** and (*S*)-(+)-6-chloroindanol ((*S*)-**1**). After 4 days at 100 µg/mL, they reduced fungus growth by 90%, 72%, and 58%, respectively. *R*-Configuration isomers exhibited slightly lower activity: at the same concentration (*R*)-**1** and (*R*)-**2** reduced fungal growth by 48% and 55%, respectively. Ketone **3** was more active than alcohol **1** but in the case of ketone **4**, the enantiopure alcohols exhibited greater antifungal activity.

These results indicate that, at least for this class of compounds, the *S*-enantiomer has a more pronounced antifungal effect. Moreover, the relative position of the chloro group in the aromatic moiety was found to be of importance since the 5-chloroindanol derivatives were more active than the 6-chloro derivatives, except in the case of ketone **3**. The results are presented in Figure 1 and Figure 2.

### 2.4. Study of the Detoxification by Botrytis cinerea UCA992

In light of the antifungal activity exhibited by the tested chloroindanol derivatives against the phytopathogenic fungus *B. cinerea*, the biotransformation of racemic alcohols **1** and **2** by *B. cinerea* UCA992 was studied to investigate a possible detoxification mechanism of the fungus.

#### Biotransformation of 6-Chloroindanol (**1**) and 5-Chloroindanol (**2**) by *Botrytis cinerea* UCA992

Biotransformation was performed in static cultures using Roux bottles for six days, after which the mycelium was filtered, extracted with ethyl acetate, and the extract dried over anhydrous sodium sulfate. The purification of the organic dry extract afforded several compounds (see Scheme 3 and Scheme 4).

In addition to the starting material, the biotransformation of 6-chloroindanol (**1**) by *B. cinerea* UCA992 afforded the known compound 6-chloroindanone (**3**), and three new compounds: *anti*-(+)-6-chloroindan-1,2-diol (*anti*-(+)-**7**), *anti*-(+)-5-chloroindan-1,3-diol (*anti*-(+)-**8**) and *syn*-(-)-5-chloroindan-1,3-diol (*syn*-(-)-**8**). The starting material recovered exhibited [α]${}_{D}^{26}$: −16.7° (*c* 0.2, CHCl_3_), 24% *ee*), in favor of configuration *R*. Based on the results of the antifungal assays, this result appears to indicate that the *S*-enantiomer is detoxified first because the *S*-enantiomer is more active against *B. cinerea* UCA992.

The chemical structures and configurations of the new compounds were determined by NMR spectroscopic studies and comparison with known compounds.

Compound *anti*-**7** was isolated as a white solid with a molecular formula of C_9_H_9_O_2_Cl as deduced from the HRESIMS analysis (*m/z* 183.0213 [M-1]^+^. Its spectroscopic data (Appendix A) were compared with those found in the literature for its diastereoisomer *syn*-**7** [26]. The main difference between the ^1^H NMR spectrum of *anti*-**7** and that of the previously described *syn*-**7** was the signal pattern of H-1 and H-2. Compound *anti*-**7** shows signals at 4.98 ppm (*d*, H-1, *J* = 5.9 Hz) and 4.38 ppm (*ddd*, H-2, *J* = 7.3, 7.3, and 5.9 Hz) corresponding to an anti-position. This hypothesis was confirmed by means of nOe’s experiments. The stereochemistry at C-1 is tentatively given as *S* configuration considering that the compound **1** recovered from the biotransformation was enriched in the *R* enantiomer. It is therefore likely that the enantiomer obtained from the biotransformation has an *S* configuration at C-1. The stereoselectivity of the reaction leading to diol **7** was very high, affording only one diastereoisomer with good enantioselectivity (73% *ee*). Compound *anti*-(+)-6-chloroindan-1,2-diol (*anti*-(+)-**7**) is reported here for the first time.

Compound **8** was obtained as a mixture of diastereoisomers with 97% *de* in favor of *syn*-configuration. Compounds *anti*-**8** and *syn*-**8** were isolated as white solids with the molecular formula C_9_H_9_O_2_Cl determined from their HRESIMS and corroborated by ^13^C NMR data (Appendix A). The NMR data were close to those of **7** but a marked lowfield shift for H-1 and H-3 in a single signal (δ_H_ 5.04 ppm for anti-**8** and 5.39 ppm for *syn*-**8**) indicated that hydroxyl groups were at C-1 and C-3. The main difference between *anti*- or *syn*- is observed for H-2 (Appendix A). Whereas H-2 and H-2′ are different signals in *anti*-**8** (δ_H_ 2.90 ppm and 1.90 ppm, respectively), both H-2 for *syn*-diastereoisomer are shown as a single signal (δ_H_ 2.37 ppm), indicating the symmetry in the molecule. Their relative configuration was also determined by nOe’s experiments. Stereochemistry at C-3 was also tentatively given as *S* configuration as it was for the compound *anti*-(+)-**7**. In this case, owing to the position of the chlorine atom, C-3 would be the equivalent carbon to C-1 in compound **7**. Compounds *anti*-(+)-5-chloroindan-1,3-diol (*anti*-(+)-**8**) and *syn*-(-)-5-chloroindan-1,3-diol (*syn*-(-)-**8**) are described here for the first time.

The biotransformation of 5-chloroindanol (**2**) by *B. cinerea* UCA992 afforded the known compounds 5-chloroindanone (**4**) and 5-chloro-3-hydroxyindanone (**10**) [27], and three new compounds: *anti*-(+)-5-chloroindan-1,2-diol (*anti*-(+)-**9**), *anti*-(+)-5-chloroindan-1,3-diol (*anti*-(+)-**8**), and *syn*-(+)-5-chloroindan-1,3-diol (*syn*-(+)-**8**). The starting material recovered exhibited [α]${}_{D}^{26}$: −27.2° (*c* 0.2, CHCl_3_), 55% *ee*), in favor of the configuration *R*.

Compound **9** was isolated as a white solid with a molecular formula of C_9_H_9_O_2_Cl deduced from its HRESIMS and corroborated by ^13^C NMR data (Appendix A). Its spectroscopic data (Appendix A) were compared with those of *anti*-**7**, only showing differences in the signal pattern for aromatic protons. Compound **9** was obtained only as a diastereoisomer in favor of the *anti*-configuration. Compound *anti*-(+)-5-chloroindan-1,2-diol (*anti*-(+)-**9**) is reported here for the first time.

Diol **8** was obtained as two diastereoisomers in favor of the *syn*-configuration (24% *de*). The spectroscopic data of both diastereoisomers corresponded to the products obtained from the biotransformation of 6-chloroindanol (**1**). Comparison of their specific rotation values showed that *anti*-**8** was the same enantiomer for both biotransformations (*anti*-(+)-**8**), but the enantiomer isolated from the biotransformation of 5-chloroindanol (**2**) was the opposite (*syn*-(+)-**8**, [α]${}_{D}^{26}$: +9.1°) of that obtained from the biotransformation of **1** (*syn*-(-)-**8**, [α]${}_{D}^{26}$:−14.1°). This result is based on the hypothesis that compounds obtained from biotransformation should have an *S* configuration at C-1.

Metabolite **10** was established as 5-chloro-3-hydroxyindanone (**10**), a known compound, by comparing its spectroscopic data with data found in the literature [27].

It is worth noting that the biotransformation of alcohol **2** afforded more products and at a higher yield than the biotransformation of substrate **1**. It was also observed that the yield of recovered starting material was higher for compound **1**, indicating that a high percentage of **2** is transformed. In addition, the mainly biotransformed enantiomer showed an *S*-configuration. This result is in accordance with the greater antifungal activity exhibited by *S*-**2** against *B. cinerea* UCA992, indicating a possible detoxification mechanism of the fungus. The main detoxification reactions involved oxidation and hydroxylation at several positions of the starting material, obtaining the corresponding ketone at C-1 and dihydroxylated derivatives.

The antifungal activity of the biotransformation products against the fungus *B. cinerea* was studied and proved to be less active than the substrates, indicating that this fungus uses oxidation reactions as a part of a detoxification mechanism. This could suggest that it is not likely to persist in the environment for long periods post-application, which is very important to improve agricultural productivity.

## 3. Materials and Methods

### 3.1. General Experimental Procedures

Optical rotations were determined with a Perkin Elmer 341 polarimeter. IR spectra were recorded on an FT-IR spectrophotometer, Perkin Elmer, Spectrum BX FTIR System. ^1^H and ^13^C NMR measurements were obtained on Agilent-400MHz, Agilent-500MHz, and Agilent-600MHz NMR spectrometers with SiMe_4_ as the internal reference. High-resolution mass spectrometry (HRMS) was performed using a Q-TOF mass spectrometer in the positive-ion ESI mode. HPLC was performed with a Hitachi/Merck L-6270 apparatus equipped with a UV-VIS detector (L 6200) and a differential refractometer detector (RI-71). TLC was performed on Merck Kiesegel 60 F254, 0.2 mm thick. Silica gel (Merck) was used for column chromatography. Purification by means of HPLC was done with a silica gel column (Hibar 60, 7 m, 1 cm wide, 25 cm long). Chemicals were provided by Fluka (Buchs, Switzerland) and Aldrich (Sigma-Aldrich, Darmstadt, Germany). All solvents used were freshly distilled. Baker’s yeast was obtained from a local shop. The following enzymes were used in this work: *Candida rugosa* Lipase (Sigma, Darmstadt, Germany, Type VII, 950 U/mg) and Porcine Pancreas Lipase (Sigma, Darmstadt, Germany, Type II) (Sigma-Aldrich, Darmstadt, Germany). Enantiomeric excesses were determined by means of HPLC analyses on a chiral column (Chiralcel OD, Daicel, Japan, 254 nm) with *n*-hexane/*i*-PrOH 95:5 as eluent, flow rate 0.8 mL/min.

### 3.2. Chemical Transformations

#### 3.2.1. Synthesis of Racemic Substrates

Compounds 6-chloroindanone (**3**) (490 mg, 0.0029 mol) or 5-chloroindanone (**4**) (500 mg, 0.003 mol) were treated with NaBH_4_ (200 mg, 0.005 mol) in methylene chloride:methanol 1:1 (200 mL). The reaction mixture was stirred for 24 h at room temperature. Distillation under reduced pressure, to eliminate the solvent, leading to a crude mixture that was neutralized with aqueous HCl 10% and extracted with ethyl acetate. The organic layer was dried over Na_2_SO_4_ and the solvent was eliminated by means of distillation under reduced pressure. The reduction mixture was chromatographed on a silica gel column eluting with hexane-ethyl acetate mixtures to give (±)-6-chloroindanol (**1**) (475 mg, 96%) and (±)-5-chloroindanol (**2**) (451 mg, 91%), respectively. The ^1^H NMR spectra of these products coincided with those found in the literature [6,9,19].

After purification, compound **1** (100 mg, 0.6 mmol) or **2** (43 mg, 0.25 mmol) were dissolved in dry pyridine (catalytic amount) and acetic anhydride (4 mL or 1.5 mL) was added dropwise. The reaction mixtures were stirred for 24 h. The solvent was then removed and the crude reaction product chromatographed to give 6-chloroindanyl acetate (**5**) (77%) and 5-chloroindanyl acetate (**6**) (89%), respectively.

**6-chloroindanyl acetate (5).** Obtained as a colourless oil. IR ν_max_ (cm^−1^): 2929, 1738, 1615, 884, 809. ^1^H-NMR (400 MHz, CDCl_3_): δ (ppm) 7.38 (d, 1H, *J* = 2.0 Hz, H-7), 7.21 (dd, 1H, *J* = 8.1 Hz, 2.0 Hz, H-5), 7.18 (d, 1H, *J* = 8.1 Hz, H-4), 6.14 (dd, 1H, *J* = 7.0, 4.0 Hz, H-1), 3.06 (ddd, 1H, *J* = 15.2, 9.0, 6.1 Hz, H-3), 2.83 (ddd, 1H, *J* = 15.2, 9.3, 5.8 Hz, H-3′), 2.52 (m, 1H, H-2), 2.11 (m, 1H, H-2′), 2.07 (s, 3H, -O-CO-CH_3_).

**5-chloroindanyl acetate (6).** Obtained as a colourless oil. IR ν_max_ (cm^−1^): 2975, 1736, 1603, 880, 818. ^1^H-NMR (400 MHz, CDCl_3_): δ (ppm) 7.33 (d, 1H, *J* = 8.1 Hz, H-7), 7.25 (bs, 1H, H-4), 7.19 (bd, 1H, *J* = 8.1 Hz, H-6), 6.13 (dd, 1H, *J* = 7.1, 3.7 Hz, H-1), 3.08 (ddd, 1H, *J* = 13.6, 7.8, 7.8 Hz, H-3), 2.86 (ddd, 1H, *J* = 13.6, 8.7, 4.7 Hz, H-3′), 2.50 (m, 1H, H-2), 2.11 (m, 1H, H-2′), 2.05 (s, 3H, -O-CO-CH_3_).

#### 3.2.2. Chemical Hydrolysis of (*R*)-(-)-6-Chloroindanyl Acetate ((**R**)-**5**) and (*R*)-(-)-5-Chloroindanyl Acetate ((*R*)-**6**)

Treatment of acetate derivatives (*R*)-**5** (111 mg, 0.53 mmol) or (*R*)-**6** (100 mg, 0.48 mmol) with KOH (46 mg, 0.8 mmol) in a methanol solution (35 mL) at room temperature and stirred for 2 h afforded enantiopure alcohol derivatives (*R*)-6-chloroindanol ((*R*)-**1**) (81 mg, 91%) and (*R*)-5-chloroindanol ((*R*)-**2**) (65 mg, 73%), respectively. (*R*)-**1**: [α]${}_{D}^{26}$ = −43°(*c* 1, CHCl_3_): >99% *ee* [6]; (*R*)-**2**: [α]${}_{D}^{26}$ = −46.5° (*c* 1, CHCl_3_): >99% *ee* [9].

### 3.3. Baker’s Yeast Transformations

A mixture composed of baker’s yeast (250 g), D-glucose (100 g), and water (1 L) was stirred in a 2 L beaker at 50 °C for 30 min. The substrate (5.93 mmol), **3** or **4**, was dissolved in a minimum amount of ethanol and then added dropwise. At the end of the reaction period, 1 L of ethyl acetate was added and the crude reaction mixture was filtered through a large Büchner funnel on a Celite pad which was later washed with the same solvent. The aqueous phase was extracted twice with 0.5 L of ethyl acetate, the organic phase was dried over anhydrous sodium sulfate (Na_2_SO_4_) and the solvent was then evaporated under reduced pressure to dryness. The residue obtained was purified by means of column chromatography.

### 3.4. Lipase-Mediated Acetylations

A mixture of the racemic alcohols **1** or **2** (100 mg, 0.6 mmol), lipase (100 mg), and vinyl acetate (5 mL) in *tert*-butylmethyl ether (12 mL) was stirred at room temperature. The residue obtained upon evaporation of the filtered reaction mixture was chromatographed on a silica gel column and eluted with hexane:ethyl acetate (95:5). The first eluted fractions afforded the acetate derivative and the last eluted fractions afforded the unreacted starting material. Detailed results of the enzyme-mediated acetylations are reported in Table 1.

### 3.5. Microorganism Culture and Antifungal Assays

The *B. cinerea* culture employed in this work, *B. cinerea* UCA 992, was obtained from Domecq vineyard grapes, Jerez de la Frontera, Cádiz, Spain. This *B. cinerea* culture has been integrated into the Mycological Herbarium Collection (Universidad de Cádiz).

The fungicide properties of chloroindanol derivatives against *B. cinerea* were established using the ‘poisoned food’ technique [25]. Bioassays entailed measuring radial growth inhibition on an agar medium in a Petri dish in the presence of test compounds at 25 °C. The test compound was dissolved in ethanol giving a final compound concentration of 50–200 µg/mL. The final ethanol concentration was identical in control and treated cultures. The medium was poured in 9 cm diameter sterile Petri dishes, and a 5 mm diameter mycelia disk of *B. cinerea* cut from an actively growing culture was placed in the center of the agar plate. Radial growth was measured for four days. Three independent experiments and three replicates per treatment were conducted. The fungicide dichlofluanid was used as a standard for comparison in this test.

### 3.6. Biotransformation of 6-Chloroindanol (1) and 5-Chloroindanol (2) by B. cinerea UCA992

*B. cinerea* UCA992 was grown at 25 °C as a surface culture on a Czapeck-Dox medium (150 mL per Roux bottle) comprised of (per liter of distilled water), glucose (40 g), yeast extract (1 g), potassium dihydrogen phosphate (5 g), sodium nitrate (2 g), magnesium sulfate (0.5 g), ferrous sulfate (10 mg) and zinc sulfate (5 mg). Substrates **1** and **2** were dissolved in ethanol and then distributed over 20 Roux bottles after 3 days of growth (100 µg/mL per bottle). Fermentation continued for a further 6 days, after which time the mycelium was filtered and then washed with brine and ethyl acetate. The broth was extracted three times with ethyl acetate and the extract was dried over anhydrous sodium sulfate. The solvent was then evaporated and the residue was chromatographed, first on a silica gel column and then with HPLC.

The following compounds were isolated from **1**: 6-chloroindanone (**3**) (15 mg, 4%), recovered (*R*)-6-chloroindanol ((*R*)-**1**) (153.9 mg, 38%) ([α]${}_{D}^{26}$:−16.7° (*c* 0.2, CHCl_3_), 24% *ee*), *anti*-(+)-6-chloroindan-1,2-diol (*anti*-(+)-**7**) (1.4 mg, 0.45%) ([α]${}_{D}^{26}$:+7.1° (*c* 0.1, CHCl_3_), 100% *de*, 73% *ee*), *anti*-(+)-5-chloroindan-1,3-diol (*anti*-(+)-**8**) (1 mg, 0.32%) ([α]${}_{D}^{26}$:+10.2° (*c* 0.1, CHCl_3_, 53% *ee*), and *syn*-(-)-5-chloroindan-1,3-diol (*syn*-(-)-**8**) (6.2 mg, 2%) ([α]${}_{D}^{26}$:−14.1° (*c* 0.6, CHCl_3_), 97% *de*, 38% *ee*).

***anti*****-(+)-6-chloroindan-1,2-diol (*anti*-(+)-7).** Obtained as a white solid, mp 154−156 °C. [α]${}_{D}^{26}$:+7.1° (*c* 0.1, CHCl_3_). IR ν_max_ (cm^−1^): 3300, 2929, 1470, 1285, 1073, 1051, 897, 827. ^1^H-NMR (600 MHz, CDCl_3_): δ 7.34 (s, 1H, H-7), 7.23 (bd, 1H, *J* = 8.1 Hz, H-5), 7.12 (d, 1H, *J* = 8.1 Hz, H-4), 4.98 (d, 1H, *J* = 5.9 Hz, H-1), 4.38 (ddd, 1H, *J* = 7.3 Hz, 7.3 Hz, 5.9 Hz, H-2), 3.22 (dd, 1H, *J* = 15.4 Hz, 7.3 Hz, H-3), 2.76 (dd, 1H, *J* = 15.4 Hz, 7.3 Hz, H-3′). ^13^C-NMR (150 MHz, CDCl_3_): δ 37.3 (t, C-3), 81.6 (d, C-2), 81.8 (d, C-1), 124.4 (d, C-7), 126.2 (d, C-4), 128.8 (d, C-5), 133.0 (s, C-6), 137.1 (s, C-9), 143.6 (s, C-8). HRESIMS *m/z*: calcd. for C_9_H_9_O_2_Cl: 183.0213 [M-1]^+^; found: 183.0210 [M-1]^+^. HPLC (Chiralcel OD, Daicel, Japan, hexane/IPA 95:5, 0.8 mL/min): *t_R_* 25.9 min (minor); 27.2 min (major).

***anti*****-(+)-5-chloroindan-1,3-diol (*anti*-(+)-8).** Obtained as a white solid, mp 164–166 °C. [α]${}_{D}^{26}$:+10.2° (*c* 0.1, CHCl_3_). IR ν_max_ (cm^−1^): 3300, 2930, 1471, 1286, 1073, 1051, 897, 827. ^1^H-NMR (600 MHz, CDCl_3_): δ 7.43 (bs, 1H, H-7), 7.37 (bd, 1H, *J* = 8.0 Hz, H-4), 7.32 (bd, 1H, *J* = 8.0 Hz, H-5), 5.04 (t, 2H, *J* = 5.8 Hz, H-1, H-3), 2.90 (dt, 1H, *J* = 13.6 Hz, 6.6 Hz, H-2), 1.90 (dt, 1H, *J* = 13.6 Hz, 5.2 Hz, H-2′). ^13^C-NMR (150 MHz, CDCl_3_): δ 46.8 (t, C-2), 72.7 (d, C-1 or C-3), 72.8 (d, C-1 or C-3), 124.7 (d, C-7), 126.6 (d, C-5), 129.2 (d, C-4), 134.8 (s, C-6), 142.7 (s, C-9), 146.2 (s, C-8). HRESIMS *m/z*: calcd. for C_9_H_9_O_2_Cl: 207.0189 [M + Na]^+^; found: 207.0181 [M + Na]^+^. HPLC (Chiralcel OD, Daicel, Japan, hexane/IPA 95:5, 0.8 mL/min): *t_R_* 26.7 min (minor); 30.0 min (major).

***syn*****-(-)-5-chloroindan-1,3-diol (*syn*-(-)-8).** Obtained as a white solid, mp 183–185 °C. [α]${}_{D}^{26}$:14.1° (*c* 0.6, CHCl_3_). IR ν_max_ (cm^−1^): 3300, 2930, 1471, 1286, 1073, 1051, 897, 827. ^1^H-NMR (400 MHz, CDCl_3_): δ 7.40 (bs, 1H, H-7), 7.35 (d, 1H, *J* = 8.1 Hz, H-4), 7.32 (bd, 1H, *J* = 8.1 Hz, H-5), 5.39 (dd, 2H, *J* = 6.2 Hz, 5.1 Hz, H-1, H-3), 2.37 (ddd, 2H, *J* = 10.3 Hz, 6.2 Hz, 5.9 Hz, H-2). ^13^C-NMR (100 MHz, CDCl_3_): δ 46.8 (t, C-2), 73.6 (d, C-1 or C-3), 73.9 (d, C-1 or C-3), 124.9 (d, C-7), 125.9 (d, C-5), 129.3 (d, C-4), 134.9 (s, C-6), 142.8 (s, C-9), 146.5 (s, C-8). HRESIMS *m/z*: calcd. for C_9_H_9_O_2_Cl: 207.0189 [M + Na]^+^; found: 207.0179 [M + Na]^+^. HPLC (Chiralcel OD, Daicel, Japan, hexane/IPA 95:5, 0.8 mL/min): *t_R_* 26.3 min (major); 39.2 min (minor).

In addition, the following compounds were isolated using **2** as a substrate: 5-chloroindanone (**4**) (5 mg, 1%), recovered (*R*)-5-chloroindanol ((R)-**2**) (45.0 mg, 10%) ([α]${}_{D}^{26}$:-27.2° (*c* 0.2, CHCl_3_), 55% *ee*), *anti*-(+)-5-chloroindan-1,2-diol (*anti*-(+)-**9**) (42.3 mg, 10%) ([α]${}_{D}^{26}$:+4.2° (*c* 0.1, CHCl_3_), 100% *de*, 37% *ee*), *anti*-(+)-5-chloroindan-1,3-diol (*anti*-(+)-**8**) (33.4 mg, 8%) ([α]${}_{D}^{26}$:+20.1° (*c* 0.2, CHCl_3_, 79% *ee*), *syn*-(+)-5-chloroindan-1,3-diol (*syn*-(+)-**8**) (54.2 mg, 13%) ([α]${}_{D}^{26}$ +9.1° (*c* 0.4, CHCl_3_, 24% *de*, 24% *ee*), and 5-chloro-3-hydroxyindanone (**10**) (2.3 mg, 0.84%) ([α]${}_{D}^{26}$:+59.7° (*c* 0.1, CHCl_3_, 55% *ee*).

***anti*****-(+)-5-chloroindan-1,2-diol (*anti*-(+)-9).** Obtained as a white solid, mp 160–162°C. [α]${}_{D}^{26}$:+4.2° (*c* 0.1, CHCl_3_). IR ν_max_ (cm^−1^): 3300, 2929, 1470, 1285, 1073, 1051, 897, 827. ^1^H-RMN (500 MHz, CDCl_3_): δ 7.35 (d, 1H, *J* = 7.8 Hz, H-7), 7.25 (d, 1H, *J* = 7.8 Hz, H-6), 7.23 (bs, 1H, H-4), 4.99 (d, 1H, *J* = 5.0 Hz, H-1), 4.54 (m, 1H, H-2), 3.12 (dd, 1H, *J* = 16.5, 5.7 Hz, H-3), 2.95 (dd, 1H, *J* = 16.5, 3.5 Hz, H-3′). ^13^C-RMN (125 MHz, CDCl_3_): δ 38.6 (t, C-3), 73.5 (d, C-2), 75.4 (d, C-1), 125.6 (d, C-4), 126.2 (d, C-7), 127.5 (d, C-6), 134.6 (s, C-5), 140.5 (s, C-9), 142.0 (s, C-8). HRESIMS *m/z*: calcd. for C_9_H_9_O_2_Cl: 183.0213 [M-1]^+^; found: 183.0730 [M-1]^+^. HPLC (Chiralcel OD, Daicel, Japan, hexane/IPA 95:5, 0.8 mL/min): *t_R_* 25.8 min (minor); 27.4 min (major).

***anti*****-(+)-5-chloroindan-1,3-diol (*anti*-(+)-8).** [α]${}_{D}^{26}$:+20.1° (*c* 0.1, CHCl_3_). HPLC (Chiralcel OD, Daicel, Japan, hexane/IPA 95:5, 0.8 mL/min): *t_R_* 26.7 min (minor); 30.0 min (major).

***syn*****-(+)-5-chloroindan-1,3-diol (*syn*-(+)-8).** [α]${}_{D}^{26}$:+9.1° (*c* 0.4, CHCl_3_). HPLC (Chiralcel OD, Daicel, Japan, hexane/IPA 95:5, 0.8 mL/min): *t_R_* 26.3 min (minor); 39.2 min (major).

**5-chloro-3-hydroxyindanone (10)** [27]. [α]${}_{D}^{26}$:+59.7° (*c* 0.1, CHCl_3_, 55% *ee*). HPLC (Chiralcel OD, Daicel, Japan, hexane/IPA 95:5, 0.8 mL/min): *t_R_* 8.4 min (major); 9.6 min (minor).

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
