# Peer review of "Biocatalytic Preparation of Chloroindanol Derivatives. Antifungal Activity and Detoxification by the Phytopathogenic Fungus Botrytis cinerea"

_plants, 2020, doi:10.3390/plants9121648_

Round 1
Reviewer 1 Report
The authors should explain why they used just B. cinerea, and why only one species and one strain was used. Indeed, the chemicals could be tested against several species to obtain a wider scenario.chapter 3.5: report: the temperature of incubation for B. cinerea; the number of plates (replicates) per treatment; the percentage of dichlofluanid in Euparen; Euparen's manufacturer; describe statistical analysisFigures 1 and 2: report statistical analysis; y-axis must not exceed 100%; dichlofluanid should be included.
Strength: the novelty of the research, results are straightforward, investigation on the mode of action of the chemicals
Weakness: only B. cinerea was tested. The chemicals could be tested against several species to obtain a wider scenario and more sound results.
Author Response
Following the very insightful and helpful comments of the reviewer we have carried out all the suggested modifications to our manuscript. Please, find below all the answers to the comments:
Botrytis cinerea UCA992 was used because it is a wild strain obtained from Domecq vineyard grapes in Jerez, and it is responsable for serious economic losses in the Jerez-Xèrés-Sherry and Manzanilla-Sanlúcar de Barrameda D.O., in southern Spain. There is a considerable need to develop fungicides to combat this phytopathogenic fungus.
Chapter 3.5: Temperatura of incubation for B. cinerea was 25ºC and three replicates per experiment were done (these data have been included in the text). Dichlofluanid 100% was used as standard fungicide and it was provided from Aldrich. Dichlofluanid is the active substance of the fungicide Euparen.
Bioassays stadistical analyses have been modified including dichlofluanid and are included as supplementary material.
Reviewer 2 Report
Par. 2; line 4: "...acetate (5) and 6-chloroindanyl acetate (6)" should be "... acetate (5) and 5-chloroindanyl acetate (6). " Table 1: The meaning of the numbers in the last column (E) should be reported
Table 1: What is the meaning of the note called b? For the optical rotations reported in the experimental section, the symbol ° appears to me with a dash; please, insert only ° Par. 3.5: it is not clear how the solutions were prepared. I guess a mother solution was prepared with ethanol as a solvent and the final ones were obtained by dilution with water. What is the concentration of the mother solutions? More details of the HPLC method must be given (instrumental details, detector, etc); moreover, from page 8 it seems that HPLC was performed in gradient elution (...with HPLC with an increasing gradient of ethyl acetate to hexane) but in the lines below it seems the elution was isocratic (hexane/IPA 95:5) Where appears, please change "hexano" to "hexane"
Author Response
Following the very insightful and helpful comments of the reviewer we have carried out all the suggested modifications to our manuscript. Please, find below all the answers to the comments:
Table 1: The Enantiomeric Ratio (E) is a selectivity factor used often in biocatalyzed kinetic resolutions. E values above 100 are considered excellent.
Table 1, Note b: The theoretical yield of each enantiomer in a kinetic resolution is limited to 50%. So, in Table 1, we consider 50% of conversion as 100%.
Par 3.5: Solutions were prepared with ethanol in all the cases, not water. The final ethanol concentration was identical in control and treated cultures.
Details of the HPLC method were given in Materials and Methods (Sec. 3.1)
Page 8, line 294: HPLC was performed in isocratic elution. We have modified the sentence deleting “with an increasing gradient of ethyl acetate to hexane”
We have corrected all the suggested typographical errors in our manuscript.
Reviewer 3 Report
The work entitled “Biocatalytic preparation of chloroindanol derivatives. Antifungal activity and detoxification by the phytopathogenic fungus Botrytis cinerea”, presented by Prof. Josefina Aleu and collaborators describes the enantiomerically pure preparation of various chloroindanol derivatives and the results of their inhibitory activity against the phytopathogenic fungus Botrytis cinerea. Likewise, they present results of the detoxification products of this fungus and it is proposed that B. cinerea uses oxidation reactions as detoxification mechanisms against some of the chlorindanols tested.
Since the work provides important information for the possible control of the phytopathogenic fungus B. cinerea, I consider it suitable for publication in the MDPI Journal Plants. The work is well written, the experimental development is adequate, and it is well described. The authors have used an adequate methodology both in the chemical part and in the biological tests. The references presented are adequate. I consider that the paper fulfills the requisites to be published in Plants and therefore I recommend its publication without any changes. My only suggestion to the authors is to share the spectroscopic data of the new compounds as supplementary material.
Author Response
The spectroscopic data of the new compounds are included as supplementary material